# Vanillic Acid as a Promising Xanthine Oxidase Inhibitor: Extraction from *Amomum villosum* Lour and Biocompatibility Improvement via Extract Nanoemulsion

**DOI:** 10.3390/foods11070968

**Published:** 2022-03-27

**Authors:** Qian Zhou, Xiaoyan Li, Xiaohui Wang, Dongdong Shi, Shengao Zhang, Yuqi Yin, Hanlin Zhang, Bohao Liu, Nannan Song, Yinghua Zhang

**Affiliations:** 1Key Laboratory of Dairy Science, Ministry of Education, Northeast Agricultural University, Harbin 150030, China; zhouqian@neau.edu.cn (Q.Z.); lixiaoyan@neau.edu.cn (X.L.); xiaohuiwang2022@126.com (X.W.); zsa1220z@163.com (S.Z.); yinyuqi06100@163.com (Y.Y.); zhanghanlin202203@163.com (H.Z.); lbh000909@163.com (B.L.); songnann97@163.com (N.S.); 2Department of Food Science, Northeast Agricultural University, Harbin 150030, China; 3Feed Research Institute, Chinese Academy of Agricultural Sciences, Beijing 100081, China; stone8119@126.com

**Keywords:** gout, oxidative stress, *Amomum villosum*, vanillic acid, response surface methodology, nanoemulsion

## Abstract

Gout is an oxidative stress-related disease. Food-derived vanillic acid, a promising xanthine oxidase inhibitor, could potentially be used as a safe, supportive, and therapeutic product for gout. The extraction of vanillic acid from a classic Chinese herbal plant *Amomum villosum* with ethanol was investigated in the study. The optimum conditions were determined as extraction time of 74 min, extraction temperature of 48.36 °C, and a solid-to-liquid ratio of 1:35 g·mL^−1^ using the Box–Behnken design (BBD) of response surface methodology (RSM). The experimental extraction yield of 9.276 mg·g^−1^ matched with the theoretical value of 9.272 ± 0.011 mg·g^−1^ predicted by the model. The vanillic acid in *Amomum villosum* was determined to be 0.5450 mg·g^−1^ by high-performance liquid chromatography–diode array detection (HPLC–DAD) under the optimum extraction conditions and exhibited xanthine oxidase (XO) inhibitory activity, with the half-maximal inhibitory concentration (IC50) of 1.762 mg·mL^−1^. The nanoemulsion of *Amomum villosum* extract consists of 49.97% distilled water, 35.09% *S_mix_* (mixture of tween 80 and 95% ethanol with 2:1 ratio), and 14.94% n-octanol, with a particle size of 110.3 ± 1.9 nm. The nanoemulsion of *Amomum villosum* extract exhibited markable XO inhibitory activity, with an inhibition rate of 58.71%. The result demonstrated the potential benefit of *Amomum villosum* as an important dietary source of xanthine oxidase inhibitors for gout.

## 1. Introduction

Gout is one of the most common metabolic diseases worldwide, which is related to the overproduction and insufficient excretion of uric acid, resulting in the formation of uric acid crystals in the kidney, joints, and other tissues [1]. Gout occurs when monosodium urate monohydrate crystals and oxidative-stress-mediated inflammatory responses [2]. Xanthine oxidase (XO) generates hydrogen peroxide and superoxide during the oxidation of xanthine to uric acid, which promotes joint injury and inflammation [3]. Therefore, XO is considered one of the most promising therapeutic targets of gout [4]. Allopurinol and febuxostat have been used clinically to treat gout by reducing xanthine oxidase activity and uric acid level [5]. However, clinically available drugs have been demonstrated to be associated with some adverse reactions, including gastrointestinal distress, hypersensitivity reaction, cardiovascular events, kidney toxicity, skin rashes, and Stevens-Johnson syndrome [6]. Recently, the demand for medicinal agents with dual xanthine oxidase inhibitor (XOI) efficacy and reduced toxicity to prevent and treat gout is increasing due to the increase in the prevalence and incidence of gout.

*Amomum villosum* Lour., which belongs to the Zingiberaceae family in classic traditional Chinese herbal plant, has been widely used in the management of numerous diseases [7]. Although *Amomum villosum* with affluent biological activities has been used to treat stomach diseases and gastrointestinal diseases, its role in the treatment of gout remains unclear [8,9,10,11]. Vanillic acid, rich in *Amomum villosum*, is a hydrophilic phenolic compound, which has been proved to be potential xanthine oxidase inhibitory activity in our previous studies [12]. The polarity of phenolic compounds is the most important basis for selecting the extraction solvent because polar compounds are easily extracted by polar solvents [13]. Different solvents and extraction methods can be used to extract the naturally occurring vanillic acid in *Amomum villosum*. Solvent extraction, as the most common extraction method, is used to obtain extracts with high phenolic content from a wide range of plants [14]. Aqueous mixtures with methanol, ethanol, and acetone are commonly used as extraction solvents due to their high extraction efficiency [15]. In particular, ethanol is the most common method employed to extract phenols due to its high safety, low price, convenient source, and recyclability. The extract yield of phenolic compounds from *Amomum villosum* depends on many experimental factors such as choice of solvent, solvent composition, extraction time, extraction temperature, and solid-to-liquid ratios [16]. The optimization of the extraction conditions of *Amomum villosum* vanillic acid is vital in order to avoid a large number of expensive and time-consuming experiments [17].

Response surface methodology (RSM) is a time- and labor-saving method to model and optimize complex processes and intended procedures [18]. It has been widely used in agriculture, biology and chemistry, the food industry, and other fields, due to its advantages of fewer experiments, higher precision, and better predictive performance [19,20,21,22]. In Box–Behnken design (BBD), a type of RSM that finds the best conditions in multifactor systems, a continuous variable surface model is built, and the influencing factors between interactions are evaluated, based on which the best horizontal range is determined [23]. Compared with the traditional orthogonal experiment or one-factor-at-a-time approach, RSM is faster and more accurate, which guides large-scale industrial production and provides useful information, even from small-scale experimental data [24]. The RSM modeling can accurately find the best combination of factors and response value in the whole region; therefore, it has become an effective method to optimize processing conditions and improve product quality [21].

Molecular docking is an important technique to predict the binding site between ligand and receptor, which provides detailed information on the binding mechanism and molecular interactions with target protein in computer simulation of protein–ligand interactions [25]. With the rapid development of molecular docking technology, it has been widely used to study the biological activity and interaction of compounds [26]. Roy et al. explored the binding mechanism of isocytosine inhibitors and XO enzyme by molecular docking simulation, determining the best conformation by analyzing the interactions between isocytosine inhibitors and XO active site, and studied the combination mode of the best conformation [27]. The efficient method of molecular docking based on theoretical calculation greatly reduces the cost and time of drug development and has the advantage of high-throughput screening and providing useful information for drug design, which is effective for exploring the ligand binding to the receptor [28].

Vanillic acid has many biological activities, including antioxidant activity, antimicrobial activity, and anticancer activity; however, its application as a functional component is currently limited due to its poor water solubility and bioavailability [29]. Emulsion-based delivery systems are increasingly used to encapsulate lipophilic bioactive components. Oil-in-water (O/W) emulsions are commonly used in delivery systems, which is beneficial to increase the solubility of lipophilic bioactive ingredients within the oil phase [30]. Nanoemulsion is a macroscopically isotropic mixture of oil, water, and surfactant, which is frequently combined with a cosurfactant to form uniform spherical droplets [31]. Nanoemulsions are considered to improve the bioavailability, stability, and effective delivery of vanillic acid within the gastrointestinal tract due to the droplet size of nanoemulsions being relatively small [32].

The main objective of this study is to employ the response surface methodology to optimize extraction conditions of *Amomum villosum* vanillic acid, with the aim of evaluating the XO inhibitory activity of *Amomum villosum* vanillic acid. Moreover, an *Amomum villosum* extract nanoemulsion system is prepared to improve the stability and biocompatibility of vanillic acid. The knowledge gained from this study should facilitate the development of XOI from *Amomum villosum* natural products and promote the rational design of delivery systems to improve the bioavailability of vanillic acid and other health-promoting lipophilic components.

## 2. Materials and Methods

### 2.1. Materials and Chemicals

The powder of *Amomum villosum* fruits was purchased from Jinhuaken of Panlong Town, Yangchun City, Guangdong Province, China. The salesperson provided the sample information of *Amomum villosum* fruits powder. Xanthine (XA) and XO were purchased from Sigma-Aldrich Co. (St. Louis, MO, USA). Vanillic acid and Folin phenol were obtained from Aladdin Industrial Co. (Shanghai, China). Tween 80 used was from Tianjin Ruijinte Chemical Company Limited (Tianjin, China). N-octanol and Ethanol (95%) were purchased from Tianjin Hengxing Chemical Company Limited (Tianjin, China). All the ingredients were food grade.

### 2.2. Single-Factor Experiment on Vanillic Acid Extraction

A single-factor design was used to explore the effects of the significant variables on the extraction of *Amomum villosum* phenols. *Amomum villosum* sample (1 g) was dissolved in 95% ethanol for solvent extraction. After extraction, the extracts were centrifuged at 4000× *g* and 4 °C for 10 min. The supernatant was collected and stored at 4 °C. The input variables were the solid-to-solution ratios of ethanol (1:15, 1:20, 1:25, 1:30, and 1:35 g·mL^−1^), extraction temperatures (40, 45, 50, 55 and 60 °C) and extraction times (20, 40, 60, 80 and 100 min). The output factors were the extraction rates of the phenolic content. According to the single-factor experimental results, the maximum values in the three groups of experiments were the optimal central values for the total phenolic content. Subsequently, the effect factors were optimized by RSM.

### 2.3. Extraction Yield Determination

The total phenols content was determined by Folin–Ciocalteu method, with modifications [33]. Under alkaline conditions, positive hexavalent tungsten ions were reduced to blue positive pentavalent tungsten ions by phenolic hydroxyl groups. At a certain concentration, the absorption of the solution at 765 nm was directly proportional to the content of phenols. The method was modified for our extracts as follows: First, 1 mL of 10–50 μg·mL^−1^ gallic acid standard solution, 1 mL of Folin phenol reagent, 3 mL of prepared 75 mg·mL^−1^ Na_2_CO_3_ solution, and 5 mL of distilled water were mixed and shaken together. The absorbance was then measured at the wavelength of 765 nm after avoiding light for 2 h. A standard curve using various concentrations (10–50 μg·mL^−1^) of gallic acid stock solutions was used to detect the total phenolic content.

The extraction yield of phenols is calculated as follows:$$Y=\frac{C\times V}{M}$$
where Y and C denote the extraction yield (mg·g^−1^) and the total volume of extraction solution after constant volume, respectively. M denotes the total mass of *Amomum villosum* powder.

### 2.4. Experimental Design by RSM

The three independent parameters—namely, extraction time (X_1_: 60–100 min), extraction temperature (X_2_: 45–55 °C), and the ratio of solid to liquid (X_3_: 1:30–1:40 g·mL^−1^)—were optimized using a three-factor, three-level BBD with RSM. The experimental design involved three process variables each at three equidistant levels (−1, 0, +1), and the extraction yield (Y) was considered as the response. The codes and levels of each factor used for RSM are shown in Table 1, as the range and center point values of the three independent variables.

### 2.5. HPLC–DAD Analysis

Vanillic-acid-rich fraction was included in the phenolic extract of *Amomum villosum* and detected by high-performance liquid chromatography–diode array detection (HPLC–DAD). Chromatographic separations were carried out using a Thermo Hypersil Gold C18 (4.6 × 250 mm, 5 μm) column. Briefly, 0.1% formic acid in water (A) and methanol (B) were used as the mobile phases. The gradient conditions were as follows: 65% A and 35% B; the total run time was 15 min. The column temperature was 35 °C, the flow rate was 0.8 mL min^−1^, and the injection volume was 15 μL. The detection wavelength was 260 nm for vanillic acid. Peaks were identified by comparing the retention times and UV spectra with standards of vanillic acid. Standard vanillic acid solutions with different concentrations ranging from 10 to 160 μg·mL^−1^ were used to calculate the standard curve according to different concentrations corresponding to different peak areas.

### 2.6. Molecular Docking

AutoDock 4.2 was performed to explore the interaction and binding mode between the vanillic acid and XO [34]. The 3D structure of XO (PDB code: 1N5X) and ligand were obtained from the RCSB Protein Database Bank (http://www.rcsb.org/pdb, accessed on: 15 August 2021) and Chem3D v18.1, respectively [35]. To obtain a stable conformation of the protein molecule, the water molecules were removed, atom types were assigned, and hydrogen atoms were added. The active site of XO was enclosed with a grid box (40 × 40 × 40 Å) with a grid spacing of 0.375 Å. The miscellaneous parameters were set to default, except that the number of Lamarckian genetic algorithm runs was set as 10 times. The docking results were further rendered with PyMOL for graphic display.

### 2.7. Amomum villosum Extract Nanoemulsion Preparation

The water titration method was used to construct pseudo-ternary phase diagrams to obtain the appropriate components, and the concentration ranges that led to a large existence area of nanoemulsion were chosen [36]. Saturated solubility with different oils (n-octyl alcohol, ethyl butyrate, ethyl acetate, oleic acid, castor oil, ethyl oleate, isopropyl myristate, and soyabean oil) was studied. The weight ratio of surfactant (tween 80) to co-surfactant (95% ethanol) was selected as 3:1, 2:1, 1:1, 1:2, 1:3 to form the three-phase diagrams. The existence areas of nanoemulsions at different temperatures (25, 45, and 65 °C), different pH levels (3, 5, 7, 9), and different levels of salinity (0.17, 0.34, 0.51 mol·L^−1^ NaCl, and distilled water) were determined. A mixture of oil and surfactant/co-surfactant (*S_mix_*) was titrated with water dropwise, under magnetic stirring at 25 °C, until the mixture became turbid. The weight of water was recorded, and the concentrations of components were calculated in percent. The pseudo-ternary phase diagram was plotted by using Origin 8.0.

### 2.8. Particle Size Measurement

The particle size of nanoemulsion droplets was measured by dynamic light scattering using a nano-ZS nanosize analyzer (Zetasizer Nano ZS90, Malvern, Worcestershire, UK) at a detector angle of 90° and a temperature of 25 °C [37].

### 2.9. Determination of XO Inhibition Assay

According to the literature [38], the XO inhibitory activity was measured at 295 nm, with minor modifications. Briefly, 50 μL of sample solutions, 60 μL of phosphate-buffered solution (0.075 mol·L^−1^, pH 7.5), and 30 μL of XO solution (0.1 U·mL^−1^) were added to a 96-well plate, respectively, and preincubated in a microplate reader at room temperature for 15 min. Subsequently, 60 μL of XA substrate solution (0.05 mol·L^−1^) was added to start the reaction at room temperature for 5 min. The XO inhibition ratio (IR) was calculated according to the following formula:$$\mathrm{IR}\left(\%\right)=\frac{\left[A_{\mathrm{control}}-\left(A_{i}{-A}_{j}\right)\right]}{A_{\mathrm{control}}}\times 100$$
where A_control_ denotes the change in absorbance without the sample, A_i_ denotes the change in absorbance with the sample, and A_j_ denotes the change in absorbance without the sample and XO. The IC50 was calculated by SPSS with concentration as abscissa and XO inhibition rate as ordinate.

## 3. Results and Discussion

### 3.1. Optimization of Single-Factor Experimental Extraction Conditions

The extraction efficiency can be affected by many factors, which directly influence the effectiveness of extraction and also engage with each other during the extraction process. The effects of the solid-to-liquid ratio, extraction temperature, and extraction time on vanillic acid from *Amomum villosum* were explored. The total phenolic content was detected with the standard curve (y = 0.0135x − 0.0116, R^2^ = 0.9979), using various concentrations (10–50 μg·mL^−1^) of gallic acid stock solutions.

The influences of solid-to-liquid ratios, extraction temperature, and extraction time on the extraction yield of phenols from *Amomum villosum* are shown in Figure 1A–C, respectively. As shown in Figure 1A, the contact area between the liquid and solid was the key factor indicating that the solid-to-liquid ratio affected the extraction of phenolic. It is obvious that the solid-to-liquid ratio exhibited a remarkably significant effect on the extraction yield. The extraction yield increased when the solid-to-liquid ratio changed from 1:15 g·mL^−1^ to 1:35 g·mL^−1^, and a slowly declining trend appeared when the solid-to-liquid ratio exceeded 1:35 g·mL^−1^. This result demonstrated that the extraction yield increased rapidly with the enhancement of the ratio of solid-to-liquid, as the *Amomum villosum* powder was completely dispersed in the ethanol solution with a larger contact area. However, with the continuously increasing volume of ethanol, the diffusion of phenols tended toward a state of equilibrium, and the contact area reached saturation so that the extraction rate increased slowly with the solid-to-liquid ratio. Therefore, a solid-to-liquid ratio of 1:35 g·mL^−1^ was chosen for further BBD experiments. Figure 1B shows the relationship between extraction temperature and extraction yield. Extraction temperature exerted a significant effect on the extraction yield. The extraction yield increased rapidly with the temperature changing from 40 °C to 50 °C, and a declined trend appeared when the temperature exceeded 50 °C because some phenols were decomposed under this condition. Therefore, the extraction temperature of 50 °C was chosen for further BBD experiments. It can be seen from Figure 1C that the extraction yield increased significantly with the extension of time between 20 min and 80 min, while it decreased when the time exceeded 80 min. The structure of phenols was destroyed after long-term heating extraction, which led to a decrease in extraction yield. Therefore, 80 min was determined as the best extraction time for further BBD experiments.

### 3.2. Box–Behnken Design Results

The extraction condition was optimized using BBD. Table 2 shows the extract yield of total phenolic content (actual and predicted) under various extraction conditions. The highest extract yield was recorded at 80 min, 50 °C, and 1:35 g·mL^−1^. The experimental data acquired from the BBD were evaluated through analysis of variance (ANOVA) to estimate the statistical significance of factors and interactions of conditions by F test and *p*-value (Table 3). The mean square deviation was divided by the residual mean square deviation to obtain the F value of each term. A larger F value with a lower *p*-value indicates a greater term significance, which is enough to explain the relationship between the actual and expected values [39]. The ANOVA results illustrated that the fitted model with a high F value (54.18) and low *p*-value (<0.0001) was remarkably significant. Notably, the lack of fit (F value = 3.06, *p*-value = 0.1542) was not remarkable, compared with the pure error. More specifically, the nonsignificant lack of fit showed the superior proximity of the proposed models for predicting the variations [40]. In addition, it can be seen from Table 3 that X_1_X_2_, X_1_X_3_, and X_2_X_3_ (*p*-value < 0.05) exhibit significant effects on the extraction yield of phenols. At the same time, regarding the extraction yield of phenols, the first term X_2_ (*p*-value < 0.05) showed extremely significant performance, while the second terms X_1_^2^, X_2_^2^, and X_3_^2^ (*p*-value < 0.05) also showed significant performance. The remaining terms showed nonsignificant performance (*p*-value > 0.05). In summary, all the tested factors triggered obvious changes in the extraction yield of phenols in different ways. The data in Table 3 show that the high R^2^ and R^2^_adj_ values were 98.58% and 96.76%, respectively. Therefore, with high reliability and precision, the proposed model was adequate to represent the experimental data.

### 3.3. Response Surface Optimization Results

The quadratic polynomial regression equations for the extraction time (X_1_), extraction temperature (X_2_), solid-to-liquid ratio (X_3_), and extraction yield (Y) were obtained as follows:$$Y=9.26\;-\;0.1487X_{1\;}-\;0.2125X_{2}+0.1212X_{3}\;-\;0.4075X_{1}X_{2}\;-\;0.3700X_{1}X_{3}+0.7675X_{2}X_{3}\;-1.13X_{1}^{2}\;-\;1.24X_{2}^{2}\;-\;0.5675X_{3}^{2}$$

Based on the equation above, the mutual interactions between the variables and the influence of the operating conditions on the yield of phenols were illustrated by using 3D response surface plots and 2D contours. As shown in Figure 2A–F, the surface in Figure 2A is relatively smooth, indicating that the interaction between the extraction time and extraction temperature was significant. The surfaces in Figure 2C–E are convex, upward, and steep, which indicates that the experimental conditions were selected reasonably, and the response value reached the maximum at the highest point. It can be seen from Figure 2B–D that the interactions between the extraction time and solid-to-liquid ratio, as well as the extraction temperature and solid-to-liquid ratio, were very significant. Moreover, the contour lines along the direction of extraction time and extraction temperature are denser than the solid-to-liquid ratio, which indicated that the effect of extraction time and extraction temperature on extraction yield was more significant. It can be concluded from the trends in Figure 2 that the influence of extraction time and extraction temperature on the response value was greater than that of the solid-to-liquid ratio.

The optimum extraction condition obtained by RSM were as follows: The extraction time was 74 min, the extraction temperature was 48.36 °C, and the solid-to-liquid ratio was 1:35 g·mL^−1^. Under these conditions, the theoretical extraction yield of phenols was 9.276 mg·g^−1^. In the actual experiment, the extraction process was adjusted to an extraction time of 74 min, extraction temperature of 48.5 °C, and a solid-to-liquid ratio of 1:35 g·mL^−1^. The actual extraction yield of phenols in three parallel experiments was 9.272 ± 0.011 mg·g^−1^, and the deviation between the theoretical value predicted by the model and the actual value was less than 10%. The content of vanillic acid in *Amomum villosum* extracts obtained under the optimal extraction conditions was detected by HPLC–DAD, with the standard curve (y = 0.6355x − 0.7032, R^2^ = 0.9997) under various concentrations (10–160 μg·mL^−1^) of vanillic acid standard solutions. It can be seen from Figure 3 that the retention time was 7.3 min for vanillic acid standard and the retention times of vanillic acid in *Amomum villosum* extracts were consistent with those of the vanillic acid standard. The content of vanillic acid in *Amomum villosum* extracts was 0.5450 mg·g^−1^. Choi et al. identified 14 compounds from *Amomum xanthioides* fruit, another medicinal plant in the family Zingiberaceae; the content of vanillic acid in *Amomum xanthioides* fruit was 0.0037 mg·g^−1^ [41]. Compared with the *Amomum xanthioides*, the content of vanillic acid in *Amomum villosum* was higher due to different extraction methods. The results showed that the proposed quadratic model was accurate and reliable to guide industrial design for the production of vanillic acid from *Amomum villosum*.

### 3.4. Molecular Docking and XO Inhibitory Activity

Figure 4 shows the results of molecular docking between xanthine oxidase and ligand. In this study, allopurinol, used as a positive control to effectively inhibit XO, and vanillic acid were docked with XO proteins to explore the docking mechanism and evaluate spatial interactions. In Figure 4, the four hydrogen bonds formed by the combination of receptors and ligands are indicated by red lines. The binding energy of vanillic acid and allopurinol were −6.98 kJ·mol^−1^ and −5.81 kJ·mol^−1^, respectively, which shows that vanillic acid was tightly bound to the active pocket of XO and could interact efficiently with XO. Tang et al. found that some key residues such as Leu648, Glu802, Arg880, PHE914, PHE1009, Thr1010, and Val1011 in the binding of phenolic compound salvianolic acid C and xanthine oxidase were tightly bound to inhibitors [35]. In this study, the hydrogen bonds were generated for allopurinol and vanillic acid at the same site of XO, in which the key residues Glu802, Arg880, and Thr1010 may be indispensable for the stability of docking complexes and contribute to the tight combination with the inhibitor. Moreover, there were hydrophobic interactions between ligands and XO, because some hydrophobic residues (Phe1009 and Phe914) in XO active pocket were adjacent to allopurinol and vanillic acid. Zhou’s research emphasized the importance of hydrophobic interactions and hydrogen bonds for the formation of stable complex conformation between compounds and XO [42]. The results obtained in this study also demonstrated that hydrogen bonds and hydrophobic interactions play key roles in maintaining the stability of the ligand-receptor docking conformation. In order to explore the potential biological activity function of *Amomum villosum* extract and demonstrate molecular docking results, the in vitro IC50 value of *Amomum villosum* extract to XO was calculated as 1.762 mg·mL^−1^, indicating the existence of active ligands for XO in *Amomum villosum*. The IC50 value of allopurinol, used as a positive control, was 1.855 μg·mL^−1^ under the assay condition. The experimental result showed good agreement, with the relatively high total scores generated by the docking analysis. These results suggested that *Amomum villosum* contains vanillic acid with XO inhibitory activity, in addition to having many phenols and flavonoids with XO inhibitory activity, which can be used as potential sources of xanthine oxidase inhibitor.

### 3.5. Amomum villosum Extract Nanoemulsion Preparation

The pseudo-ternary diagrams constructed are shown in Figure 5, the areas of the nanoemulsion region and isotropic regions on the diagram increased. Under the conditions of *S_mix_* = 2:1, pH of 7, the temperature of 25 °C, distilled water as the water phase, and n-octanol as the oil phase, the maximum nanoemulsion region was observed, hence selected for further studies of XO inhibitory activity. Figure 6 illustrates the particle size distributions of the optimum nanoemulsion and vanillic acid extract nanoemulsion. The particle sizes of the obtained optimum nanoemulsion and vanillic acid extract nanoemulsion were determined to be 110.3 ± 1.9 nm and 116.2 ± 2.2 nm, respectively. It can be seen from the particle size distributions that the diameters of oil droplets in the two nanoemulsions were unimodal, which further indicated that nanoemulsions reached a monodispersed stable system and could effectively transport substances. The inhibition rates of 20 μg·mL^−1^
*Amomum villosum* extract nanoemulsion and *Amomum villosum* extracts were 58.71% and 51.99%, respectively, owing to tween 80 acting as a hydrophilic stabilizer and protecting the emulsion better [43]. Moreover, vanillic acid extract tends to be more lipophilic, which makes the overall oily phase linked with the surfactants at the oily phase side [44]. Therefore, the ability to inhibit XO in *Amomum villosum* extract nanoemulsions and vanillic acid extracts is similar at the same concentration. The results further illustrated that the process used in the preparation of *Amomum villosum* extract nanoemulsion would not damage vanillic acid, and the various auxiliary materials used would not affect the activity of vanillic acid in inhibiting XO. The formulation of *Amomum villosum* extract nanoemulsions, which composed of 49.97% distilled water, 35.09% *S_mix_* (mixture of tween 80% and 95% ethanol with 2:1 ratio), and 14.94% n-octanol, could be an appropriate selection. Overall, the nanoemulsion provided an effective and stable delivery system for transporting vanillic acid or other bioactive components.

## 4. Conclusions

In this study, response surface methodology, combined with a Box–Behnken design, was successfully employed to optimize extraction from *Amomum villosum* using solvent extraction. The optimum extraction conditions were acquired. The extraction time was the most influential factor due to its significant impact on the level of extraction yield of phenols. Additionally, *Amomum villosum* extract showed strong and effective xanthine oxidase inhibitory activity, which can be used as a source of xanthine oxidase inhibitor. An optimal nanoemulsion formulation was obtained. Furthermore, owing to the fact that the *Amomum villosum* nanoemulsion improved the solubility of vanillic acid and effectively protected the XO inhibitory activity of vanillic acid, the XO inhibitory activity of *Amomum villosum* nanoemulsion was higher than that of vanillic acid. This study will be beneficial to the potential application of vanillic acid extracted from *Amomum villosum* in developing functional foods for patients suffering from gout.

## Figures and Tables

**Figure 1 foods-11-00968-f001:**
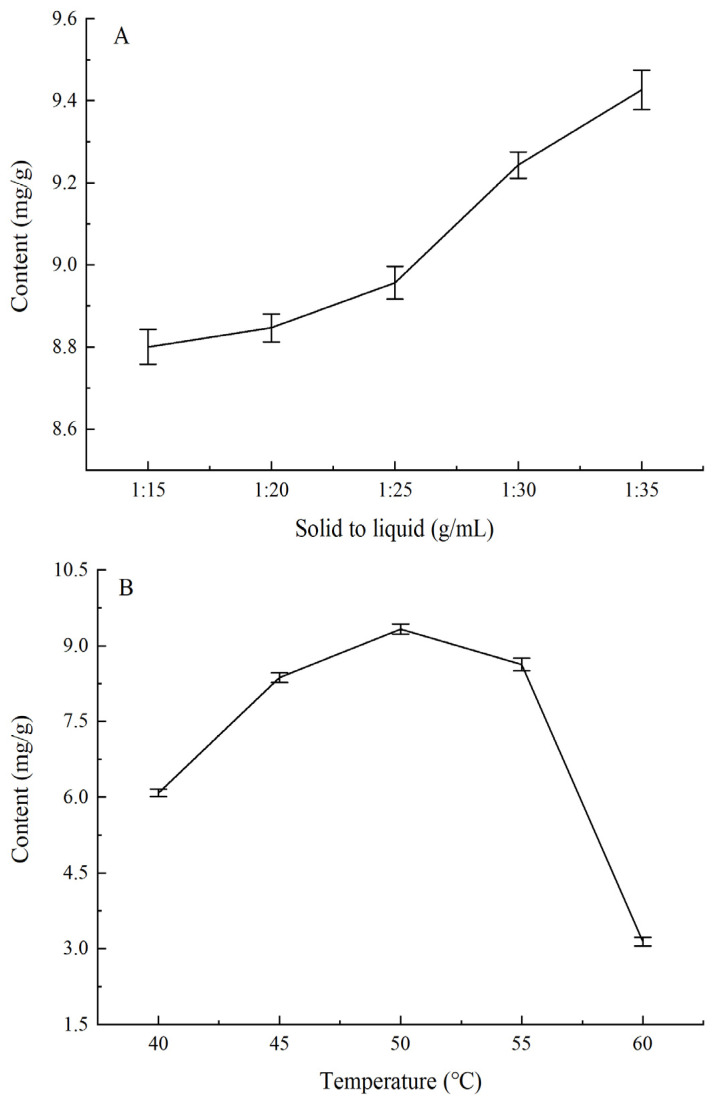
Effects of (**A**) the solid-to-liquid ratio, (**B**) extraction temperature, and (**C**) extraction time on the extraction of phenols from *Amomum villosum*.

**Figure 2 foods-11-00968-f002:**
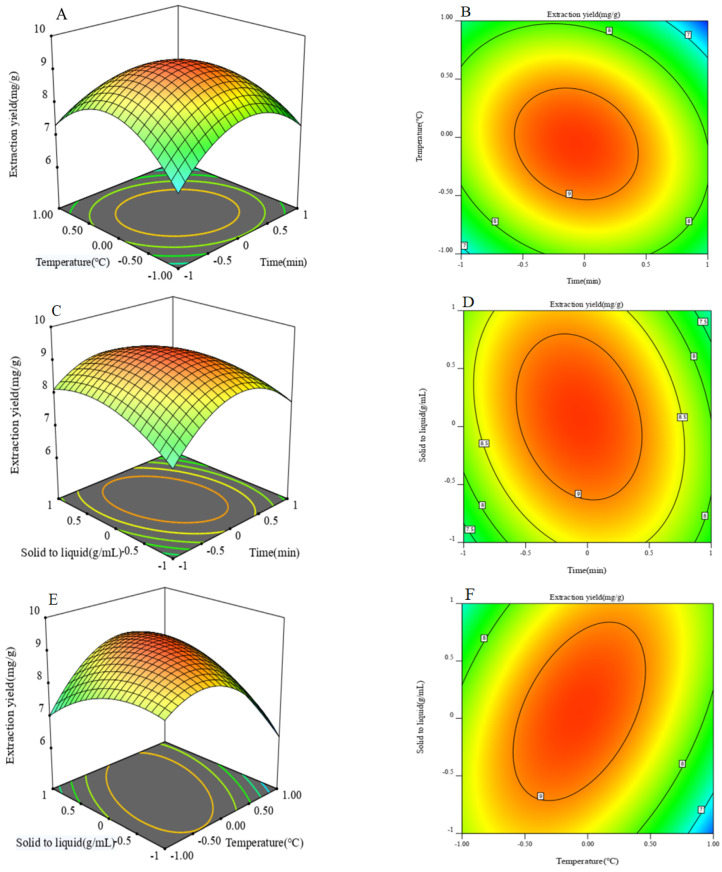
Response surface plots of the effects of factor interactions on phenols extraction yield: (**A**) Response surface of interaction between extraction temperature and extraction time; (**B**) The contour of interaction between extraction temperature and extraction time; (**C**) Response surface of interaction between the solid-to-liquid ratio and extraction time; (**D**) The contour of interaction between the solid-to-liquid ratio and extraction time; (**E**) Response surface of interaction between the solid-to-liquid ratio and extraction temperature; (**F**) The contour of interaction between the solid-to-liquid ratio and extraction temperature.

**Figure 3 foods-11-00968-f003:**
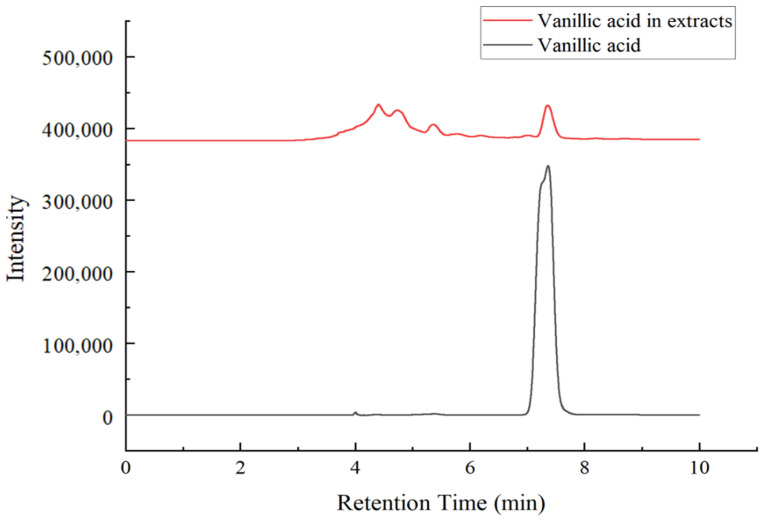
The qualitative and quantitative results of *Amomum villosum* vanillic acid by high-performance liquid chromatography–diode array detection.

**Figure 4 foods-11-00968-f004:**
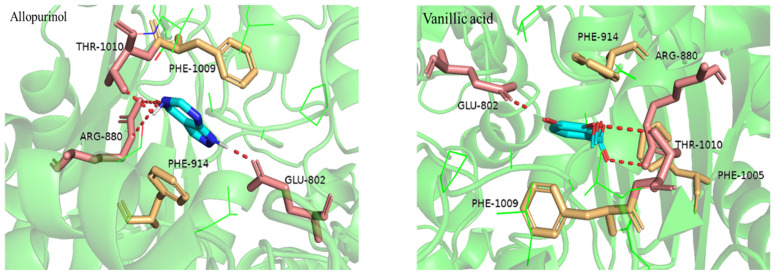
Xanthine oxidase and ligand molecular docking diagram.

**Figure 5 foods-11-00968-f005:**
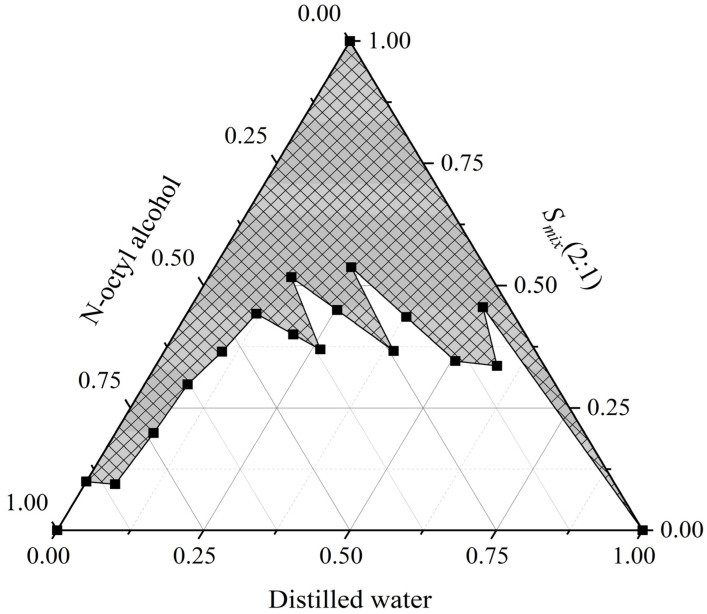
The optimum construction of *Amomum villosum* extract nanoemulsion containing water, oil, Tween 80%, and 95% ethanol, with 2:1 ratio.

**Figure 6 foods-11-00968-f006:**
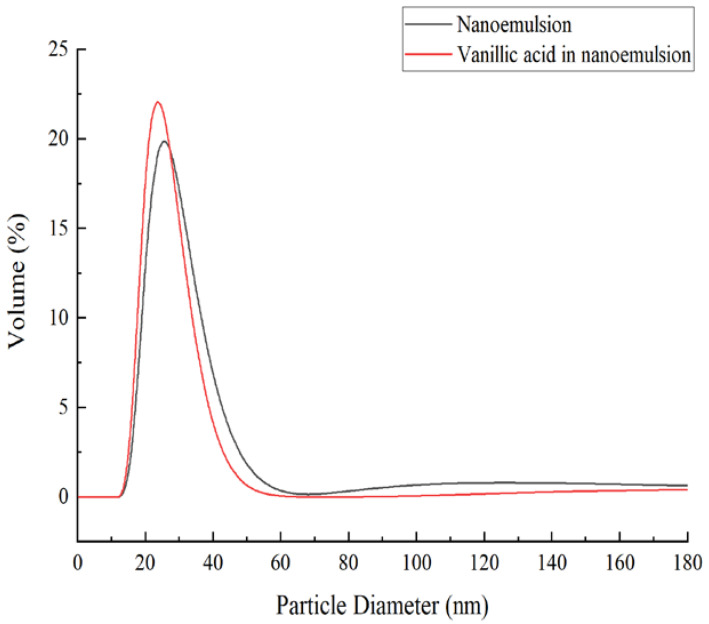
Particle size distributions for nanoemulsion and *Amomum villosum* extract nanoemulsion.

**Table 1 foods-11-00968-t001:** The coded levels of extraction time (X_1_), extraction temperature (X_2_), and solid-to-liquid ratio (X_3_).

Levels	Minimum Point (−1)	Central Point (0)	Maximum Point (+1)
X_1_ (min)	60	80	100
X_2_ (°C)	45	50	55
X_3_ (g·mL^−1^)	1:30	1:35	1:40

**Table 2 foods-11-00968-t002:** Three-level factorial design and results of extraction time (X_1_), extraction temperature (X_2_), and ratio of solid to liquid (X_3_).

Number	X_1_	X_2_	X_3_	Actual (mg·g^−1^)	Predicted (mg·g^−1^)
1	1	0	1	7.1	7.17
2	0	−1	1	6.89	7.02
3	0	0	0	9.45	9.26
4	1	0	−1	7.61	7.67
5	1	1	0	6.05	6.13
6	0	0	0	9.21	9.26
7	−1	0	1	8.26	8.21
8	−1	1	0	7.04	7.24
9	−1	−1	0	6.93	6.85
10	0	0	0	9.35	9.26
11	−1	0	−1	7.29	7.22
12	0	−1	−1	8.17	8.32
13	0	0	0	9.21	9.26
14	0	0	0	9.08	9.26
15	1	−1	0	7.57	7.37
16	0	1	−1	6.49	6.36
17	0	1	1	8.28	8.13

**Table 3 foods-11-00968-t003:** Analysis of variance (ANOVA) and interactions for the total phenol content (Y) from *Amomum villosum* extracts as a function of extraction time (X_1_), extraction temperature (X_2_), and ratio of solid to liquid (X_3_).

Source	Sum of Square	Degree of Freedom	Mean Square	F Value	*p*-Value Probability
Model	18.72	9	2.08	54.18	<0.0001
X_1_	0.1770	1	0.1770	4.61	0.0689
X_2_	0.3612	1	0.3612	9.41	0.0181
X_3_	0.1176	1	0.1176	3.06	0.1235
X_1_X_2_	0.6642	1	0.6642	17.30	0.0042
X_1_X_3_	0.5476	1	0.5476	14.26	0.0069
X_2_X_3_	2.36	1	2.36	61.38	0.0001
X_1_^2^	5.35	1	5.35	139.43	<0.0001
X_2_^2^	6.42	1	6.42	167.29	<0.0001
X_3_^2^	1.36	1	1.36	35.32	0.0006
Residual	0.2687	7	0.0384		
Lack of Fit	0.1871	3	0.0624	3.06	0.1542
Pure Error	0.0816	4	0.0204		
Total	18.99	16			
R^2^ = 98.58%					
R^2^_adj_ = 96.76%					

## Data Availability

Authors confirm that the study did not report any data.

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
