# Peer review of "Vanillic Acid as a Promising Xanthine Oxidase Inhibitor: Extraction from Amomum villosum Lour and Biocompatibility Improvement via Extract Nanoemulsion"

_foods, 2022, doi:10.3390/foods11070968_

Round 1

Reviewer 1 Report

GENERAL COMMENT

            The authors discussed an interesting topic in the manuscript (ms) which makes their interesting results regarding the vanillic acid extraction from Wurfbainia villosa also known as Amomum villosum with concomitant nanoemulsion production more accessible. However, there is no a complete discussion about the statistical design applied to reach the optimal extract condition according with other yield researches to enhance your interesting results. Moreover, the reviewer suggests a major revision including grammar revision by specialized institution.

Major concerns
Comment 1: Abstract needs revision, in two main regards.  First, the authors must name not only the kind of RSM used, i.e. Box-Behnken design (BBD), but also the solvent used during optimization process.  Second, results must be better explored and presented, such as the proportions of the components of the nanoemulsion as well as the particle diameter.

Comment 2: Introduction must include the importance of using the molecular docking on your research.

Comment 3: Material and methods: Please describe the IC50 calculation by SPSS at item 2.9.

Comment 4: Results and discussion: Please provide a better explanation about the evaluated parameters during the optimization process by BBD as well as their interactions described on your ms. About the content of vanillic acid in your research, it is possible to compare com other recent studies for the same specie or another family member? It is mandatory to cite all the missing references during discussion. Table 3 must be better explored during discussion.

Comment 5: Figures and Tables must be self-explained. Thus, all conditions and abbreviations must be completely written. Moreover, each Table and Figure must mention the tested specie.

  Specific comments

Please use the correct way to write the scientific name of species in your ms, even in Title and references. Also, may you verify if the tested specie was Wurfbainia villosa or Amomum villosum?

L49-55, 221. Please avoid the redundant information in such lines.

L72-74, Item 2.4, 2,6. Please place the missing references.

Please correct the grammar mistakes and the correct use of subscripts in the ms.

Tables 2 and 3: What X1, X2 and X3 means?

Figures 3 and 4 legends must be more descriptive, what XO means?

Figure 5. What Smix means?

Figure 6 legend must not describe the main finding.

Reviewer 2 Report

Comments

The author of the manuscript entitled “Vanillic Acid as a Promising Xanthine Oxidase Inhibitor: Extraction from Amomum Villosum Lour and Biocompatibility Improvement via Extract Nanoemulsion” is a good work and important for the scientific community. There are some errors as mentioned in the comments below:

Title: Change “Amomum Villosum Lour” into “Amomum villosum Lour.”

Line 14,18,23, 24,42: keep “Amomum Villosum” in italic form throughout the text

Define abbreviated form in the abstracts

HPLC-DAD, IC50, XO inhibitory

Keywords: avoid abbreviation in keyword list

RSM

Line 44: Make new para

“Amomum villosum Lour., which belongs to the Zingiberaceae family in classic…..”

Line 165: check the spelling

“was studyed.”

Line 188: check

“Here, Acontrol denotes……”

Write the units in the following format

μg/mL replace by μgmL-1

 Line 313: define XOI in the following

“which can be used as a potential source of XOI……“

Discussion

Discuss the result by citing recent and related work. Mechanism by which vanillic acid can be used as Xanthine Oxidase Inhibitor is completely missing.

Conclusion

Avoid using data in the conclusion part of the manuscript.

Reviewer 3 Report

In the MS entiteled "Vanillic Acid as a Promising Xanthine Oxidase Inhibitor: Ex-2 traction from Amomum Villosum Lour and Biocompatibility 3 Improvement via Extract Nanoemulsion"

the authors used factorial design to find optimum conditions for extraction of vannilic acid then put in nanoemulsion and worked on XO enzyme beside of docking study.

The MS sounds good still there are minor comments.

  1. I could not find the chemical analysis e.g nmr, mass or any tool to analyze the extracted compound to be vannillic acid.
  2. the authors wrote in line 21 vannilic acid extract: what do you mean by the word extract? did you do your biological investigation using pure vannilic acid and prepare your nanoemulsion from pure vannilic acid or combination of phenolic compounds including major vannilic acid rich fraction??? as from the title i concluded the authors work on pure vannilic acid compound. kindly explain this point.
  3. the chinese herb named "Amomum Villosum Lour" should be written in italic all over the MS except for Lour i.e Amoum villosum Lour since this is a latin botanical name.
  4. You mentioned in line 97: the authors purchased the herb but they did not mention which organ? is it the herb, leaves, or what part of the plant? also there is no authentication of the powder or any prepared voucher specimen prepared for the plant?
  5. In line 97: Yangchun: what is this? place in china or name of herbal store and what is the source of the plant powder? did the authors purchased from local store??
  6. in line 140: Vanillic acid, the phenolic extract of A. villosum: what does this sentence mean? either the authors isolate vannilic acid pure or phenolic rich fraction enriched with vannilic acid? kindly explain.
